# The Influence of Fiber on the Mechanical Properties of Controllable Low-Strength Materials

**DOI:** 10.3390/ma16155287

**Published:** 2023-07-27

**Authors:** Yafeng Qian, Mingyang Jiang

**Affiliations:** School of Architecture and Transportation, Liaoning University of Technology, Fuxin 123000, China

**Keywords:** CLSM, fiber, compressive strength, splitting strength

## Abstract

Numerous studies have been conducted on fiber-reinforced concrete; however, comparative investigations specifically focusing on the utilization of fibers in CLSM remain limited. In this study, we conducted a systematic investigation into the mechanical properties of controlled low-strength material (CLSM) by manipulating the length and doping amount of fibers as control variables. The 7-day compressive strength (7d-UCS), 28-day compressive strength (28d-UCS), and 28-day splitting strength of CLSM were employed as indicators to evaluate the material’s performance. Based on our comprehensive analysis, the following conclusions were drawn: (1) A positive correlation was observed between fiber length and material strength within the range of 0–6 mm, while conversely, a negative correlation was evident. Similarly, when the fiber doping was within the range of 0–0.3%, a positive correlation was identified between material strength and fiber doping. However, the strength of CLSM decreased when fiber doping exceeded 0.3%. (2) SEM and PCAS analyses provided further confirmation that the incorporation of fibers effectively reduced the porosity of the material by filling internal pores and interacting with hydration products, thereby forming a mesh structure. Overall, this study offers valuable insights into the manipulation of fiber length and doping amount to optimize the mechanical properties of CLSM. The findings have important implications for the practical application of CLSM, particularly in terms of enhancing its strength through fiber incorporation.

## 1. Introduction

Up to the present, the cumulative stock of global red mud is about 7 billion tons, but the comprehensive utilization rate is less than 5% [1], and a large quantity of red mud storage seriously pollutes the natural environment [2]. Tailings are solid waste generated after beneficiation, and the annual discharge of iron ore tailings is increasing, posing a great threat to the natural environment [3]. Therefore, how to properly dispose of these solid wastes has become an urgent issue to be solved [4,5].

In recent years, there has been more and more research on the use of industrial solid waste CLSM such as red mud tailing sand [6]. Cases of applying CLSM to pavement base layer are also endless, including the pavement backfill [7] and pavement base layer [8,9], but it has been found in many studies that CLSM applied to the pavement base layer, like traditional concrete, has the disadvantages of high brittleness and low tensile strength. Therefore, solving the problem of concrete cracking has become an important issue. In order to address the issue of highway pavements prone to cracking under persistent heavy loads, extensive research has been conducted by both domestic and foreign scholars. Studies have shown [10,11] that adding various fibers, including glass and polypropylene, to concrete can greatly improve its birth defects. These studies have demonstrated that the addition of fibers to concrete can effectively prevent cracking and improve the fatigue life of concrete. Specifically, BF (abbreviation for a specific type of fiber) has been found to possess high tensile strength, leading to significant enhancements in shear and impact strength of the subgrade [12]. Fan W [13] discovered that incorporating BF into pavement materials effectively enhances the performance of semi-rigid asphalt pavement, improves its service quality, and extends its lifespan. Some scholars [14,15] have investigated the effect of the fusion of fibers from different cementitious materials on the performance of pavement subgrade. They concluded that the application of cementitious material fibers for pavement subgrade is feasible. Wang J. et al. [16] employed scanning electron microscopy (SEM) and X-ray computed tomography (CT) to examine the improvement mechanism of PAN fibers. Petrounias P. et al. [17] studied the preparation of concrete and evaluated its mechanical properties using hair fibers as ditives. Furthermore, studies [18,19,20] have demonstrated the positive effects of incorporating various fiber materials such as polypropylene and glass into road subgrade mixtures in different proportions. These studies conducted comparative analyses between mixtures with fibers and those without fibers, assessing mechanical and physical properties such as uniaxial compressive strength and flowability. The results indicate that mixtures with fibers exhibit improved mechanical and durability properties. Yanchao [21] observed enhanced road performance when GF (another type of fiber) was incorporated into GF concrete for pavement subgrade compared to the control group. These studies explored the feasibility of applying different fibers to concrete in various scenarios. Xue Z. et al. [22] determined that the optimal properties of concrete were achieved when the mixture contained 0.06% BF content with a fiber length of 18 mm. SEM analysis of fiber mixtures [23,24] revealed that the small “villi” on the fiber surface promoted better adherence to the cement mortar binder, thus enhancing the bonding and overall integrity of the fibers in the mixture. Xu G. et al. [25] discovered that the addition of PP (abbreviation for a specific type of fiber) limits crack formation in treated slurry, thereby improving the strength of hardened slurry. Rosas M. et al. [26] found that the addition of GF to concrete enhances the tensile and flexural strength of concrete. Several scholars [27,28,29] have also examined the impact of GF on concrete workability from various perspectives. Li Y. et al. [30] explored the advantages of BF when used in concrete, focusing on the study of concrete durability. Khan M. et al. [31] analyzed the microscopic relationship between BF length and content and concrete strength. Zhang C. et al. [32] studied the macroscopic micromechanical properties and reinforcement mechanism of alkali-resistant GF steel concrete in an alkaline environment. Ermolovich E. A. et al. [33] conducted microstructural and petrographic analysis of backfill materials using scanning electron microscopy.

The above-mentioned studies conducted a comprehensive evaluation of waste materials such as tailings and CLSM, and assessed the effects of different fiber materials, parameter changes, and single-factor variations on the performance of CLSM. Research on fiber materials started decades ago and has yielded significant results. Studies have been conducted using fibers such as polypropylene and glass in CLSM, but due to engineering differences, only a few studies have analyzed the influence of different fibers on the strength of CLSM. Exploring the strength development laws of different fiber types can help us understand their advantages and applicability, providing guidance for fiber selection in practical applications. By introducing different types of fibers into CLSM, a comprehensive comparative study of pavement subgrade was conducted with the aim of deeply exploring the influence of different fibers on the strength of CLSM. The results of this study will promote and enhance the application of CLSM in these specific situations.

Therefore, in our study, we refer to the research conducted by Jiang M et al. [4]. Based on the optimal ratio of CLSM prepared using red mud tailings, we introduce different fibers, including glass, basalt, and polypropylene, into the material. The fiber length and content are considered as controlled variables. The unconfined compressive strength and splitting strength of CLSM serve as the design indicators. In order to investigate the influence of different fibers on these three indices of CLSM, we employ the Taguchi design method to determine the appropriate matching ratios.

## 2. Materials and Methods

### 2.1. Iron Tailings

Iron tailings (IOT) were sourced from the Qidashan iron mine located in Anshan, Liaoning Province. The density test method refers to this study [34] (the red mud and slag test method is the same). The particles of iron tailings exhibit a finer granularity and manifest in distinct cyan and gray states. The density of the tailings was determined to be 2754 kg/m^3^. The original morphology of the tailings, as well as the morphology after screening, are illustrated in Figure 1a,b, respectively. In order to ascertain the chemical composition of the tailings, X-ray fluorescence (XRF) analysis was conducted. The primary chemical constituents of the tailings are presented in Table 1. Analysis of Table 1 reveals crucial insights regarding the elemental composition of IOT. Evidently, the dominant constituents of this sand variant are identified as Fe_2_O_3_, SiO_2_, and CaO, which account for the majority of its composition. In addition to these primary elements, trace amounts of MnO and MgO are also discernible within the sample.

### 2.2. Red Mud

Red mud (RM) is derived from the waste generated during the aluminum production process of the Chalco Shandong enterprise. It appears as a solid block with an earthy red color and has a density of 2630 kg/m^3^. The red mud appears reddish-brown, and the original red mud particles are relatively large. After crushing, the size of the red mud particles becomes relatively small, resulting in the formation of fine particles or powdery substances. To prepare the red mud for further use, the original material undergoes a drying process in an electric blower drying oven at 105 °C for 3 h. Referring to the test method in the relevant literature [34], the water content of the red mud was measured to be 8.5%. Subsequently, the dried red mud is crushed and sieved through a 0.15 mm square hole screen. The original morphology of the red mud, as well as its morphology after crushing, can be observed in Figure 2a,b respectively. After coarse grinding and fine grinding, the particle size of the red mud is approximately 300 mesh, meeting the usage requirements. The chemical composition of the red mud was determined using X-fluorescence rays (XRF) analysis. The primary chemical constituents of the red mud are outlined in Table 1. Examination of the data presented in Table 2 elucidates crucial insights into the elemental composition of red mud. Notably, the primary constituent of red mud is identified as Fe_2_O_3_, constituting a significant proportion (78.93%) of its total composition. Furthermore, SiO_2_ represents 4.99% of the sample, while TiO_2_ comprises 8.32%. In addition to these predominant components, trace amounts of MnO and Na_2_O are also discernible within the sample.

### 2.3. Slag

The slag used in this study is classified as S95 grade fine powder, which is supplied by Hebei Cangzhou Ultrafine Special Cement Company. It possesses a density of 1429 kg/m^3^. The original morphology of the slag is depicted in Figure 3. X-ray fluorescence (XRF) analysis was conducted to determine the chemical composition of the slag. The principal chemical constituents of the slag are presented in Table 1. The visual representation in Figure 3 illustrates that the slag exhibits a granular morphology, with these granules comprising fine powders. Moreover, these powders are observed to be agglomerated, forming a cohesive state. Analysis of the data presented in Table 3 reveals essential insights into the elemental composition of the slag. It is evident that the primary constituents of the slag are identified as CaO and SiO_2_, which account for 74.89% and 18.07% of the total content, respectively. Furthermore, the slag sample also contains trace amounts of K_2_O, MgO, and MnO.

### 2.4. Fibers

Glass Fiber (GF) is an inorganic, non-metallic synthetic material produced by subjecting pyrophyllite, quartz sand, limestone, dolomite, boronite, brucite, and other ores to high-temperature melting, wire drawing, winding, weaving, and other processes. It consists primarily of silica, alumina, calcium oxide, boron oxide, magnesium oxide, and sodium oxide. The density of GF ranges from 2.4 to 2.76 g/cm^3^, offering notable properties such as good insulation, strong heat resistance, excellent corrosion resistance, and high mechanical strength.

Basalt Fiber (BF) has a density range of 2.65 to 3 g/cm^3^ and is characterized by its high tensile strength. It also exhibits electrical insulation, corrosion resistance, and high-temperature resistance.

Polypropylene (PP) is an isotactic polypropylene produced through polymerization technology. The synthetic fiber made from PP using a special process has a density range of 0.87 to 0.95 g/cm^3^. It possesses a melting temperature ranging from 107 °C to 141 °C and exhibits various excellent properties, including high strength, good elasticity, wear resistance, and corrosion resistance.

For this test, the GF used is sourced from Shandong Tai’an Songze Composite Material, the BF is purchased from Shandong Tai’an Songze Composite Material, and the PP is acquired from Haining Anjie Composite Material. The specific physical properties of the different fibers are presented in Table 2. Three fiber lengths were selected for the test—3 mm, 6 mm, and 12 mm—and their morphologies are illustrated in Figure 4.

### 2.5. Test Equipment and Equipment

This section describes the types of equipment required for testing, as shown in Table 3. All equipment usage can be found in this paper [34].

### 2.6. Test Methods

Based on a sand-to-sand ratio of 0.255, a NaOH content of 3.667%, and a mass concentration of 80.657% [4], this test focuses on three representative fibers: GF, BF, and polypropylene. The variables for fiber length are set at 3 mm, 6 mm, and 12 mm, while the fiber volume content serves as the variable for this test, with levels set at 0.15%, 0.3%, and 0.45%. The experimental design employs the three-factor, three-level design method in DOE Taguchi’s design scheme. The design objective is to optimize the mixing ratio of CLSM in the rubber sand ratio by considering the 7-day unconfined compressive strength (UCS), 28-day UCS, and 28-day splitting strength of CLSM.

A brief description of the manufacturing process for the material experiment is as follows:Raw material preparation: Prepare the required tailings, red mud, slag, and fiber according to the experimental requirements. Include additives needed for testing to ensure they meet specifications and purity requirements.Mixing and preparation: Mix and prepare the weighed ingredients. Follow the test methods of materials stabilized with inorganic binders for highway engineering, which includes ingredient weighing, stirring, and mold loading.Curing: Place the prepared specimens in a standard curing box for 24 h and then demold them for further curing.

The test adopts the method of a parallel experiment, with nine groups for each of the three fibers, and each group containing 81 samples. This results in a total of 243 samples for the three fibers. If the experimental error exceeds 15%, the test pieces need to be prepared again.

In accordance with ASTM C192/C192M-19 specifications [35], CLSM specimens are cylindrical with dimensions of Φ50 mm × 100 mm. The specimen is illustrated in Figure 5a. Standard curing methods are employed, with a temperature of 20 ± 2 °C and a humidity level greater than 95%.

The unconfined compressive strength test follows the ASTM-C39/C39M-21 [36] standard. The top surface and side of the CLSM specimens are kept flat and smooth, and the WDW-100E universal testing machine is used for UCS testing.

The CLSM splitting strength test adheres to the Test Regulations for Stable Materials of Inorganic Binders in Highway Engineering [37]. The TAW-2000 geotechnical triaxial testing machine is used for this test, along with a press, splitting fixture, and gasket. The specifications of the compressive press align with those specified in the Test Regulations for Inorganic Binding Stabilized Materials. Since fine-grained iron tailings are used in this test, the sample dimensions prepared should be Φ50 mm × 50 mm, as depicted in Figure 5b.

For the microscopic investigation, test blocks that meet the SEM test standard are selected from fractured samples at each age. They are soaked in absolute ethanol to halt the hydration reaction. Before testing, the test blocks are removed and placed in a 70 °C drying oven for 3 h. A flat and smooth surface is selected for testing.

## 3. Results and Discussion

### 3.1. Design Solutions

The DOE design methodology will be used for this experiment. The DOE design method possesses the remarkable capability to effectively reduce the number of tests required, thereby shortening the overall test period. Furthermore, this method has been proven to consistently yield stable and reliable results. There were nine groups of PP, GF, and BF design schemes, for a total of 27 sets of tests. Table 4 shows the DOE design table, and Table 5 and Table 6 show the test results of CLSM unconfined compressive strength and splitting strength, respectively.

### 3.2. Analysis of Test Results

In order to make the data results more intuitive, the test results in the above table are categorized and graphically processed according to influencing factors such as fiber type and its length, as described below.

#### 3.2.1. Effect of Fiber Length on CLSM Strength

Taking the fixed fiber content of 0.3% as an example, the influence of different fiber lengths of 0 mm, 3 mm, 6 mm, and 12 mm on the splitting strength of CLSM 7d-UCS, 28d-UCS and 28d was explored. The figure below 0 shows the influence of different fiber lengths on CLSM strength.

It is evident from Figure 6 that at a constant fiber content, the incorporation of fibers has a limited impact on the compressive strength of the 7-day age sample when the fiber length is 3 mm. The strength enhancement is most pronounced with 6 mm fibers, and 12 mm fibers affect the strength of CLSM. From Figure 6a, it can be observed that when the fiber length exceeds 3 mm, the compressive strength of the 7-day specimens containing the three fibers exceeds the compressive strength of the unadulterated fiber group. The specimen with a fiber length of 6 mm exhibits the highest strength, and the 7-day compressive strength of all three fibers exceeds 5 MPa. The most effective fibers in reinforcing strength are PP and GF, followed by BF. A fiber length of 6 mm is able to form a mesh structure with the matrix material, thereby increasing the 7-day compressive strength. However, when the fiber length is 12 mm, the compressive strength of the specimen decreases. Upon analysis, it can be concluded that with shorter fiber lengths, there are fewer contact points between the fibers, making it difficult to effectively transfer stress, and the fibers are unable to provide sufficient strength support and reinforcement, leading to a reduction in the material’s strength. Fibers of moderate length can form a more supportive network within the material, effectively withstanding stress and resisting fracture. Excessively long fibers may cause fiber interlacing or fiber lamination during material processing and molding, resulting in reduced effective utilization of fibers and thereby affecting the strength of the material. From Figure 6b, it can be observed that the 28d-UCS strength of CLSM follows a similar development pattern to the 7d-UCS. It is worth noting, however, that the late strength increase in the 3 mm glass fiber is higher than that of the other two fibers. This discrepancy is presumed to be related to experimental error when combined with the analysis of the other groups.

According to the results of the referenced study [4], the 28-day splitting strength of the undoped fiber CLSM was 0.43 MPa. Figure 7 shows the different magnitude increases in the 28-day splitting strength when fibers of different lengths were incorporated into the CLSM. It is worth noting that when the fiber length is 3 mm, the 28-day splitting strength of GF reaches 0.6 MPa, which is 50% higher than that of the undoped fiber group. This increase is significantly higher than the observed increase in compressive strength. The splitting strength of PP and BF also exceeds 0.4 MPa. When the fiber length is 6 mm, the splitting strength of the three types of fibers all reach peak value. The 6 mm fiber length has the most significant improvement in the splitting strength of the substrate. For small samples prepared from fine tailings, fiber lengths that are too short or too long can compromise sample integrity. However, the length of 6 mm can maximize the strength of the substrate, which is consistent with the strength analysis results. On the other hand, a slight decrease in splitting strength was observed when the fiber length was increased to 12 mm. Among the fibers, the decrease in BF was the largest. This may be due to the accumulation of long fibers in the material, which hinders the uniform dispersion and leads to the settling of small fiber parts. The interaction between fibers will form stress concentration points, which will easily cause the occurrence and propagation of cracks. Consequently, the strength of the material is affected; thus, the splitting strength of CLSM decreases.

#### 3.2.2. Effect of Fiber Content on CLSM Strength

Taking a fixed fiber length of 6 mm as an example, the study investigated the influence of different fiber content, including 0%, 0.15%, 0.3%, and 0.45%, on the splitting strength of CLSM at 7-day, 28-day, and overall durations. 

It can be observed from Figure 8 that the fiber content has a significant impact on the 7-day unconfined compressive strength (7d-UCS) of the base material. When the fiber length is 3 mm, the strength of the three fiber types shows minimal variation compared to the control group without fiber incorporation. Moreover, the analysis indicated a significant decrease in the 7-day ultimate compressive strength (UCS) of the specimens with added PP compared to the control group. It is worth noting that this disparity could be attributed to experimental errors, which, nonetheless, remained within acceptable limits.

From Figure 8a, it can be seen that the 7d-UCS of CLSM is the highest when the fiber content is 0.3%. The strength of glass fiber and PP fiber is about 5.2 MPa, while the influence of BF is relatively weak. This can be explained as the optimal fiber content effectively combines with the material to form a dense network structure that enhances the strength of the substrate. However, when the fiber content reaches 0.45%, the strength of the substrate decreases significantly. This drop can be attributed to excess fiber volume compared to the material volume, causing some of the fibers to entangle and form clumps or sediments within the material. As a result, these unevenly distributed fibers cannot adequately bond to the material, thereby affecting its overall strength [38]. In Figure 8b, when the fiber content is in the range of 0–0.3%, the CLSM strength is proportional to the fiber content; and when the fiber content is 0.45%, the CLSM strength decreases. Among them, glass fiber has the best overall performance. To sum up, the appropriate fiber content has a significant impact on the material strength. When the fiber content is too high, the interaction between fibers may lead to a decrease in the binding force of the fiber bundles, thereby reducing the overall strength of the material. In addition, excessive fiber content may also lead to an increase in the brittleness of the material because the interaction between fibers will form stress concentration points, which will easily cause the occurrence and propagation of cracks.

By analyzing the line graph in Figure 9, it is clear that the inclusion of fibers has a significant positive effect on the splitting strength of the substrate. When the fiber doping is in the 0–0.3% interval, the splitting strength of CLSM improves with the increase in fiber doping, and the overall effect on the strength performance of glass fibers is the best within this interval. The strength of CLSM decreases when the fiber doping exceeds 0.3%, with the most significant decrease in the strength of polypropylene fibers. It is speculated that the reason for this is that since the tailing sand belongs to fine-grained soil, when the fiber doping is in the range of 0–0.3%, the included fibers are able to form a good bond with the tailing sand and other materials, allowing the fibers to effectively intertwine and combine with the tailing sand and other granular materials. This bonding enhances the strength and stability of the material and provides better mechanical properties. When the fiber doping exceeds 0.3%, the number of fibers added to the material exceeds the reasonable proportion or design requirements, and an excessive amount of fibers may lead to mutual interference and obstruction between the fibers, thus weakening the overall strength of the material. Overall, the trend in splitting strength aligns closely with the trend in compressive strength. However, upon comparing the two, it was found that the increase in splitting strength with the addition of fibers was significantly higher than the increase in fiber compressive strength.

### 3.3. Microscopic Investigation of Fibers

#### 3.3.1. Fiber CLSM Scanning Electron Microscopy (SEM) Analysis

In order to better explore the principle of fiber curing base material, the 28d-year-old fiber samples were scanned by electron microscopy, and the SEM plots of GF, BF, and PP 28d samples were shown in Figure 10.

As shown in Figure 10, the fibers are interspersed and interwoven inside the material, and the distribution of the fibers is disordered. This combination reduces the interconnection between the pores and improves the bulk density of the particles, thereby reducing the internal porosity of the material. Moreover, it can be observed that the overall morphology of the material is relatively flat, and the internal pore state is crack-like. Therefore, the increase in the material’s strength can be attributed to these factors. Figure 10a presents a scanning electron microscope (SEM) image of the GF, revealing the dispersed GF fibers throughout the material, occupying areas not filled with hydration products. This dispersion leads to the formation of a dense network structure within the material [39]. Conversely, Figure 10b displays an SEM image of BF, where the BF fibers are predominantly longitudinally distributed. It is noteworthy that when the fibers are distributed only transversely or longitudinally, their effect on increasing the material’s strength tends to be weaker compared to cross-distributed fibers. This explains why the material strength of the BF group is lower than that of the GF group. Figure 10c exhibits the SEM image of PP. In this group, the fibers are uniformly distributed, and there is high adhesion between the fibers and the material. However, closer examination reveals that only a small number of fibers are distributed transversely in the scanning area, indicating poor uniformity of fiber cross-distribution. The inhomogeneous distribution of fibers hinders further improvement in the material’s strength. Analysis of the SEM images confirms that GF exhibits the most optimal distribution within the material, followed by PP, while BF exhibits the least optimal distribution. These findings align with the results of the above data analysis.

#### 3.3.2. Pore Structure Analysis of Fiber Samples of Various Ages

The obtained SEM images were lineated and vectorized by PCAS software, and the porosity, probability entropy, probability distribution index and fractal dimension values of SEM images of samples of different ages were calculated. The following Figure 11, Figure 12 and Figure 13 show gray threshold images and recognition threshold images at GF, BF, and PP 28d ages, respectively.

As indicated in Table 7, the porosity of 28-day-old samples incorporated with different fiber CLSM is recorded as 3.33%, 3.98%, and 3.97%, respectively. These values are lower than the porosity of undoped fiber CLSM at the same age. Furthermore, the pore probability entropy of samples at each age exceeds 0.95, approaching 1. This suggests that in the fiber-doped CLSM, all material particles exhibit random orientation. The probability distribution index of different fibers hovers around 2.2. The incorporation of fibers into the material allows for the division of internal pores, resulting in the formation of smaller pores with varying specifications and a reduction in the number of larger pores. Moreover, the fibers effectively fill the areas that are not occupied by hydration products. This phenomenon serves as the primary reason why the probability distribution index of fiber-doped samples is lower than that of undoped fibers at the same age. Additionally, due to the intervention of fibers, when filling the larger holes, a certain number of smaller holes are also filled, thus maintaining a relatively consistent fractal dimension compared to the previous period.

## 4. Conclusions

In this experiment, the Taguchi design method was employed to select three representative fibers: glass fiber (GF), basalt fiber (BF), and polypropylene (PP). The fiber length variables were set to 3 mm, 6 mm, and 12 mm, while the fiber content variable levels were set at 0.15%, 0.3%, and 0.45%. The compressive strength and splitting strength were used as indicators to design the test scheme and analyze the test results. The study aimed to investigate the influence of GF, BF, and PP fibers with different lengths and contents on the strength of the base material, in addition to conducting scanning electron microscopy (SEM) analysis of the three fibers. The main findings are summarized as follows:The influence of different fiber lengths on the strength of CLSM materials can be summarized as follows: Within the range of 0–6 mm fiber length, there exists a positive correlation between fiber length and material strength, disregarding the minimal experimental error. It is worth noting that the strength of CLSM reaches its peak at a fiber length of 6 mm. However, as the fiber length is extended to 12 mm, the material strength demonstrates varying degrees of decline. Additionally, it has been observed that fibers enhance the splitting strength of CLSM to a greater extent than its compressive strength. Based on a comprehensive analysis, it can be concluded that glass fiber exhibits the most favorable overall performance effect among the tested fibers.The effects of different fiber lengths on CLSM strength can be summarized as follows: Fiber doping within the range of 0–0.3%, excluding experimental errors, exhibits a positive proportional relationship with the material strength. Notably, the material strength reaches its peak at a fiber doping of 0.3%. However, when the fiber doping increases to 0.45%, the material strength shows a decrease. Furthermore, it can be observed that fibers enhance the splitting strength of CLSM to a greater extent than its compressive strength. Among the fibers tested, 0.3% glass fiber doping demonstrates the most favorable overall performance effect. This finding highlights the potential of glass fiber as an effective reinforcement for CLSM materials.SEM analysis of the fibers reveals that they are interspersed within the material, with intertwinement and overlap between fibers. This arrangement leads to a reduction in the internal porosity of the material, which is the main contributing factor to its strength enhancement. Moreover, analysis of the pore structure of the fiber materials indicates that their porosity is lower than 0.04, confirming their ability to improve the material’s strength compared to the undoped fiber control group.In summary, the findings of this study provide clear evidence regarding the influence of various fiber lengths and contents on the strength of CLSM. Moreover, SEM analysis serves to further validate the feasibility of utilizing fibers to enhance the material strength. Therefore, this study offers valuable insights and serves as a significant reference for the application of fibers in CLSM. However, at the same time, we should consider whether the incorporation of other solid waste materials can be better applied in the pavement base and explore its application on higher-grade pavements. Additionally, the durability, fluidity, and other properties of CLSM should also be analyzed. For microscopic analysis, methods such as XRD and FTIR can be used for more in-depth analysis.

## Figures and Tables

**Figure 1 materials-16-05287-f001:**
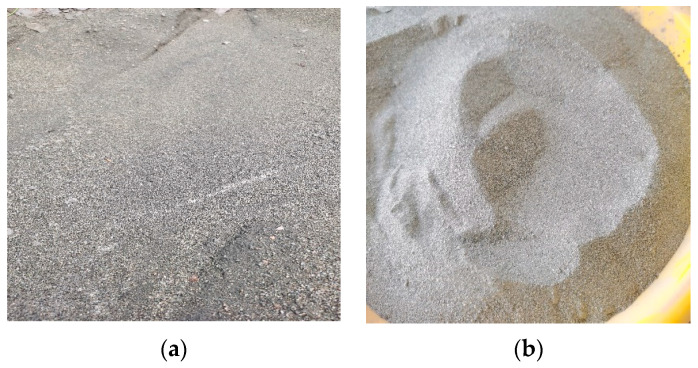
Original iron tailing sand morphology (**a**) and sieved iron tailing sand morphology (**b**).

**Figure 2 materials-16-05287-f002:**
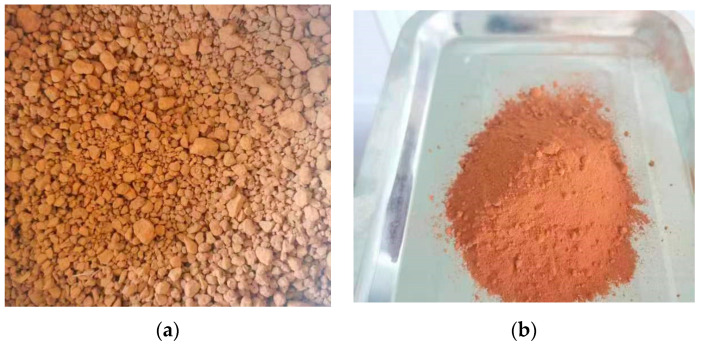
Original morphology (**a**) and post-crushing morphology (**b**) of red mud.

**Figure 3 materials-16-05287-f003:**
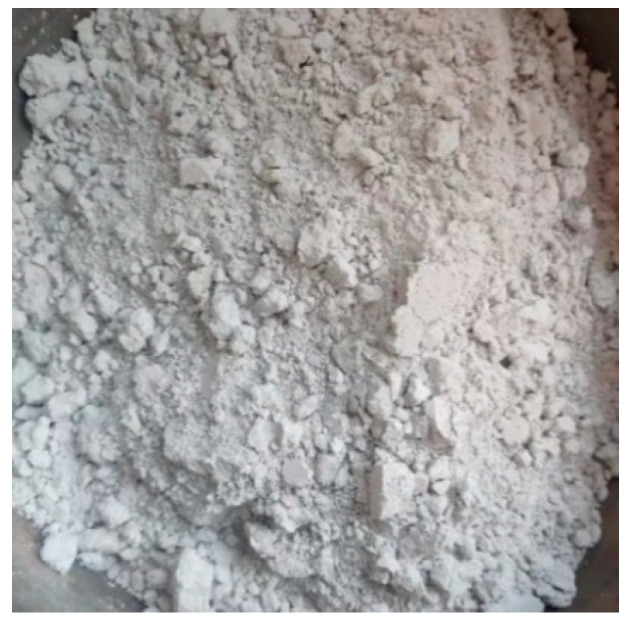
Slag morphology.

**Figure 4 materials-16-05287-f004:**
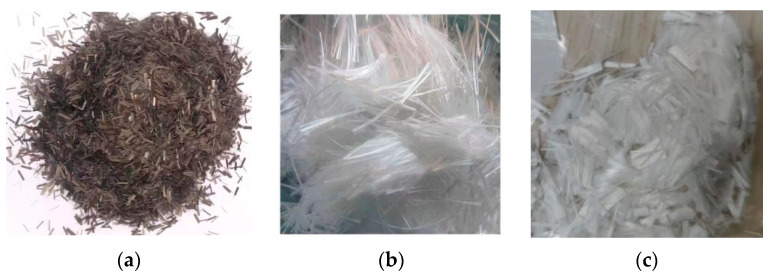
Morphology of basalt fibers (**a**), polypropylene fibers (**b**), and glass fibers (**c**).

**Figure 5 materials-16-05287-f005:**
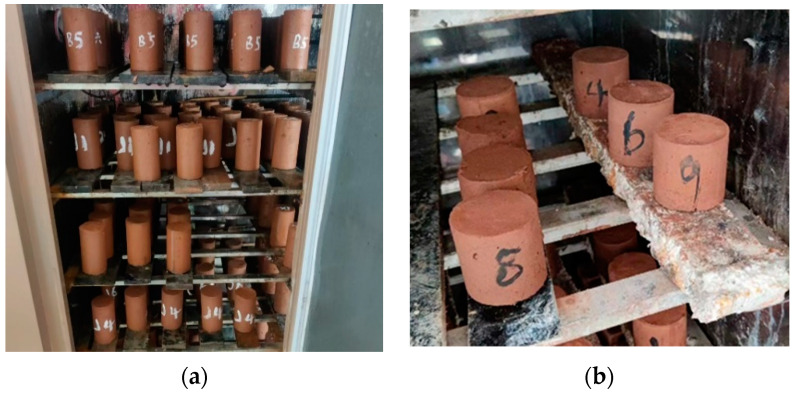
Specification diagram of CLSM unconfined compressive strength specimen (**a**) and splitting strength specimen (**b**).

**Figure 6 materials-16-05287-f006:**
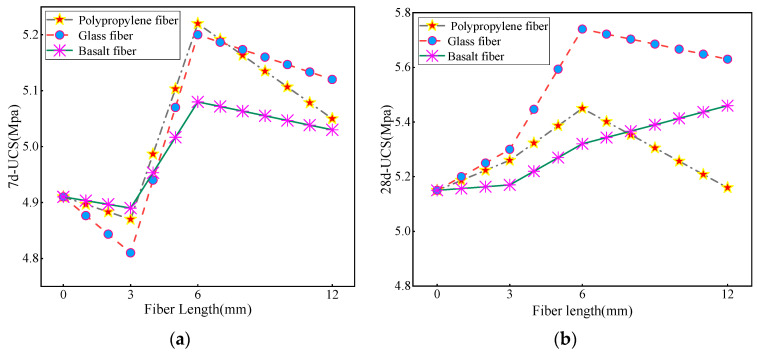
Effect of fiber length on the strength pattern of CLSM 7d-UCS (**a**) and 28d-UCS (**b**).

**Figure 7 materials-16-05287-f007:**
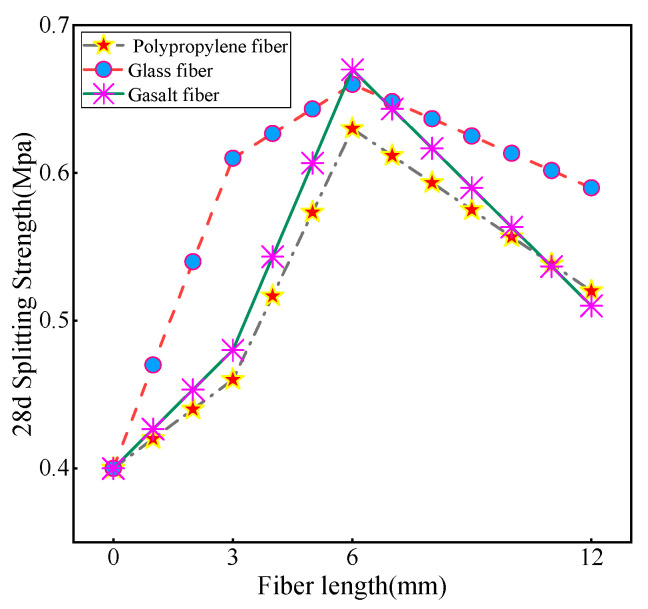
Trend of the effect of fiber length on the 28d splitting strength of CLSM.

**Figure 8 materials-16-05287-f008:**
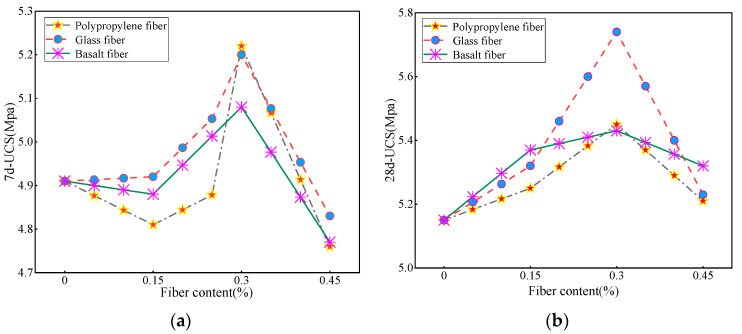
Effect pattern of fiber doping on strength of CLSM 7d-UCS (**a**) and 28d-UCS (**b**).

**Figure 9 materials-16-05287-f009:**
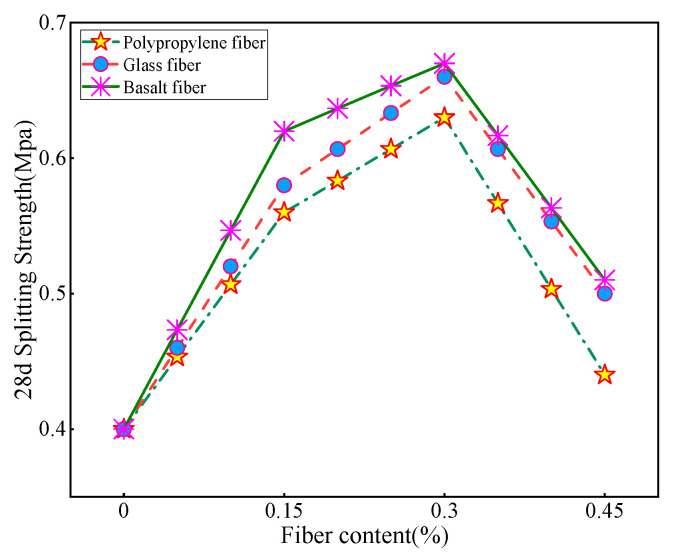
Trend of the effect of fiber dosing on the 28d splitting strength of CLSM.

**Figure 10 materials-16-05287-f010:**
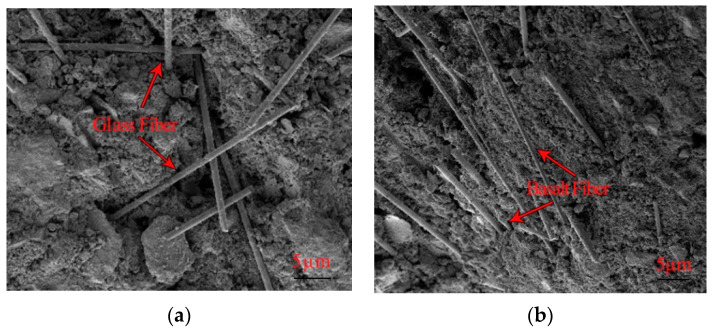
SEM images of glass fiber (**a**), basalt fiber (**b**), and polypropylene fiber (**c**) 28d samples.

**Figure 11 materials-16-05287-f011:**
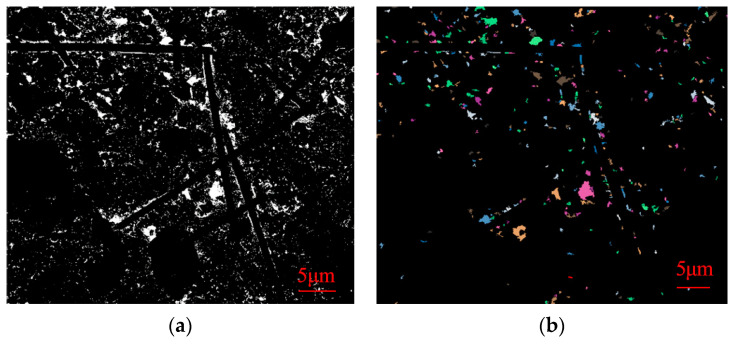
Gray threshold image (**a**) and recognition threshold image (**b**) of GF 28d age.

**Figure 12 materials-16-05287-f012:**
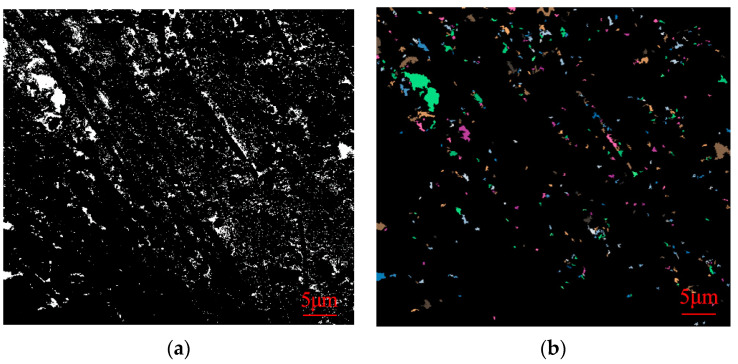
Gray threshold image (**a**) and recognition threshold image (**b**) of BF 28d age.

**Figure 13 materials-16-05287-f013:**
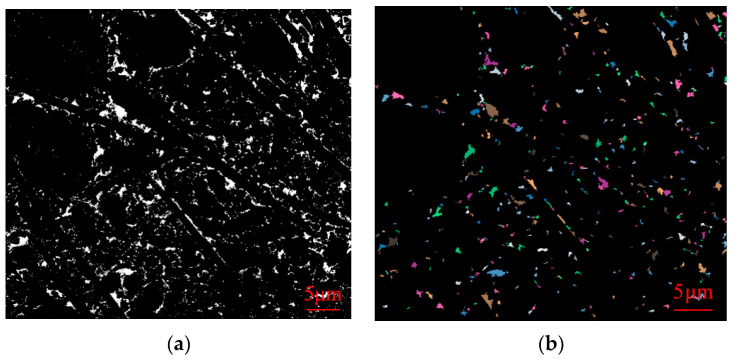
Grayscale threshold image (**a**) and recognition threshold image (**b**) at PP 28d age.

**Table 1 materials-16-05287-t001:** Chemical composition of IOT, RM, and Slag (Wt%).

	Mineral Type	Fe_2_O_3_	SiO_2_	CaO	K_2_O	Al_2_O_3_	TiO_2_	Other
Content (Wt%)								
IOT		29.84	38.72	14.83	7.33	6.80	3.71	1.77
RM		78.93	4.99	1.41	0.2	4.02	8.32	2.13
Slag		0.75	18.07	74.89	1.05	3.85	-	1.39

**Table 2 materials-16-05287-t002:** Physical property parameters of fibers.

Fiber Type	Specific Gravity g/cm^3^	Diameter/μm	Tensile Strength/MPa	Modulus of Elasticity/GPa	Elongation at Break/%
BF	2.8	18	3900	35	3.5
PP	0.91	21	633	3.5	26.5
GF	2.5	13	1800	80	2.45

**Table 3 materials-16-05287-t003:** Instruments and equipment used in the test and description.

Instrument Name	Instrument MODEL	Production Company	Test Method
Electronic scales	MJJ003	Meilen Corporation	-
Cement sand mixer	JJ-5	Bovis Corporation	-
Concrete vibrator table	HZJ-0.5	Hojo Corporation	-
Concrete curing tank	YH-90B	Shunchang Building Material Factory	-
Testing machine	WDW-100E	Jinan Times Metallurgical Testing Machine Co.	-
Triaxial testing machine	TAW-2000	Changchun Chaoyang Testing Instruments Co.	-
X-ray diffractometer	X’Pert PRO MPD	Nalytical Corporation	2*θ* 10°–80° 5°/min
Scanning electron microscope	Zeiss Sigma 300	Zeiss Corporation	Gold spray coating Acceleration voltage 30 kV
Drying oven Electric heating blast drying oven	101-3B	Soper Instruments Ltd.	Related User Manuals

**Table 4 materials-16-05287-t004:** DOE design solutions for three types of fibers: PP, GF, and BF.

Fiber Type	PP		GF		BF	
Group	Dosing (%)	Length (mm)	Dosing (%)	Length (mm)	Dosing (%)	Length (mm)
1	0.15	3	0.15	3	0.15	3
2	0.15	6	0.15	6	0.15	6
3	0.15	12	0.15	12	0.15	12
4	0.3	3	0.3	3	0.3	3
5	0.3	6	0.3	6	0.3	6
6	0.3	12	0.3	12	0.3	12
7	0.45	3	0.45	3	0.45	3
8	0.45	6	0.45	6	0.45	6
9	0.45	12	0.45	12	0.45	12

**Table 5 materials-16-05287-t005:** Unconfined compressive strength results for PP, GF, and BF fibers.

Fiber Type	PP		GF		BF	
Group	7d-UCS	28d-UCS	7d-UCS	28d-UCS	7d-UCS	28d-UCS
1	4.78	5.16	4.85	5.34	4.83	5.29
2	4.81	5.25	4.92	5.32	4.88	5.37
3	4.93	5.31	5.12	5.40	5.16	5.50
4	4.87	5.26	4.81	5.30	4.89	5.17
5	5.22	5.45	5.20	5.74	5.08	5.32
6	5.05	5.16	5.12	5.63	5.03	5.46
7	4.77	5.15	4.91	5.39	4.66	5.39
8	4.76	5.21	4.83	5.23	4.77	5.43
9	4.62	5.01	4.50	4.95	4.75	5.38

**Table 6 materials-16-05287-t006:** Results of 28-day split tensile strength of PP, GF, and BF fibers.

Fiber Type	PP	GF	BF
Group	28d	28d	28d
1	0.49	0.56	0.58
2	0.56	0.58	0.62
3	0.54	0.53	0.64
4	0.46	0.61	0.48
5	0.63	0.66	0.67
6	0.52	0.59	0.51
7	0.49	0.54	0.53
8	0.44	0.50	0.51
9	0.42	0.45	0.56

**Table 7 materials-16-05287-t007:** Pore structure parameters of samples at different ages.

Pore Structure Parameters	GF	BF	PP
Porosity	3.33%	3.98%	3.97%
Probabilistic entropy	0.9847	0.9723	0.9847
Probability distribution index	2.2852	2.2783	2.2097
Fractal dimension	1.1838	1.2051	1.1208

## Data Availability

All data that support the findings of this study are included within the article.

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
