# Peer review of "The Influence of Fiber on the Mechanical Properties of Controllable Low-Strength Materials"

_materials, 2023, doi:10.3390/ma16155287_

Round 1
Reviewer 1 Report
A major problem, which made my task as a reviewer harder, is that the authors did not insert line numbers in their text. It is logic and mandatory to do it before submission.
Title: You do not need this in the title. Start it with "The influence of fiber ........".
Abstract is ok but in the keywords, you use CLSM, PP, GF and BF. They must be given in full at least some of them or the most important one, e.g. polypropylene fiber (PP), basalt fiber (BP) and controllable low-strength material (CLSM).
Why do you superscript reference notation?. You must follow the instructions for authors strictly.
Fig. 1: Quality of iron tailing and its morphology photos are bad. Please use other ones with better resolution, brightness and contrast.
Table 1: Oxide contents should be given in wt% and not just %. Also, you need to indicate that this XRF analysis indicates total iron as ferric, i.e. Fe(III). For other minor oxide contents, it is much better to name them as a footnote or at least mention them in the text. Based of your analysis, the others are mostly MgO, MnO, Na2O and possibly others.
Fig. 2: Photos of the hand specimen sample need to be of a better quality.
For Tables 2 and 3, please consider all comments for Table 1. I suggest that authors put all XRF analyses of the different materials use in a single table and re-organize and re-number in the text.
Fig. 3: It is highly recommended to describe the morphology of any material such as slag in this case. The caption must be informative.
Fig. 4: It was much better to enhance the quality of basalt fiber photos if the view is taken when samples are outside the plastic bag.
Table 5: Some punctuation should be considered.
Tables 6, 7 and 8 should include Fiber type in the first row. Caption of the three tables can be modified in which you can start with "Results of .....".
Fig. 6: Try to improve quality of notation for x- and y-axes in this figure same as you do n Fig. 7. Also for Fig. 6 only, please provide name of each fiber in full as well as abbreviated in the key or legend.
In page 10, why do you provide the paragraph in a bold format?.
Fig. 12: The provided SEM images lack scale.
Please provide better contrasted gray thershold images in Figs. 13, 14 and 15. Do not forget to insert bar scale too.
The conclusion section is good but it is much better if you can provide it in the form of bullets. So, please modify and consider summarization as well. Of course you need to keep the main issues of your findings especially the behaviour of different fibers and their characteristics. Try to show how do type and length are the most effective as obtained from your experiments and the trends yield.
References: Please follow the instructions for authors to prepare your reference list properly.
I attach an annotated pdf with my review so that you can use for the preparation of a revised version of your manuscript.

Not bad English and needs fine polishing only.
Author Response
We thank the reviewers for their comments and responses to this study. The authors have made changes in accordance with the reviewers' comments, which are described below:
1、The title has been revised, and line numbers have been added according to the reviewer's suggestions. (P1,L1)
2、Thanks to your suggestions, the authors have revised the keywords. (P1,L21)
3、The authors have revised the citation of references according to the reviewers' instructions. (P3,L91)
4、Thanks to your reply, the authors have reshoot the iron tailings and related samples.
5、The authors have revised Table 1 (including Table 2 and Table 3) and added relevant notes according to the reviewers' comments. (P4,L121;P5,L139;P5,L143)
6、Thanks to your suggestion, the authors have added the description of tailings and red mud morphology. (P2,L83-L84;P3,L94-L96)
7、Thanks to your suggestion, the authors have retaken the fiber photos. (P5,L138)
8、The authors have modified Table 5 (changed to Table 3) according to the reviewers' comments. (P5,L143)
9、Thanks to your suggestions, the authors have revised Tables 6, 7, and 8 (which have been changed to Tables 4, 5, and 6) and the titles of the tables. (P7,L186;P8,L187;P8,L188)
10、The authors have revised the images according to the responses given by the reviewers. (P9,L196;P10,L215;P10,L233;P11,L255;)
11、The bold paragraph on page 10 was revised. (P11)
12、Thanks to the reviewer's suggestion, the authors have scaled the SEM images (including the grayscale and color threshold images below) and marked the fibers in the SEM images. The authors have found through software manipulation that the existing gray images have the best contrast, and it is no longer possible to add gray images with better contrast. (P12,L273;P13,L297;P14,L298)
13、The authors have partially revised the paper according to the reviewers' suggestions for the conclusion. (P14-P15)
14、Thank you for your reply, and the authors have prepared the reference list according to the reviewers' instructions. (P17-P18)

Reviewer 2 Report
The manuscript “Study on the influence of fiber on the mechanical properties of controllable low-strength materials” studied the influence of various type of fibers (polypropylene fiber, glass fiber and basalt fiber) on the mechanical properties of controllable low-strength materials. Τhe results are well presented. The authors have employed correctly the techniques. I have found the methodological approach correct. The presentation of the problem is clear, the results correctly presented and the conclusions well explained. English is not bad and generally is easy to follow, but there are some evident grammar mistakes, which, in most cases, do not preclude the comprehension; hence the necessity of a revision, or not, would depend on the exigency of the journal in this aspect. To conclude, I suggest this manuscript to be published in the journal “Materials” after the below major revisions:
Ø The first section introduces the problem to a reader. It is well written, concise and informative. Some references should be added in the introduction for the use of different types of fibers to increase the strength of low-strength materials.
References to be cited in the introduction field:
· An Innovative Experimental Petrographic Study of Concrete Produced by Animal Bones and Human Hair Fibers. Sustainability, 2021.
· Optimization of the compressive strength of hair fiber reinforced concrete using central composite design. IOP Conf. Ser.: Mater. Sci. Eng. 2020.
Ø A better description of the purpose should be given.
Ø Please provide the methodology descriptions of all the equipment used.
Ø It would be convenient to include a table of abbreviations.
Ø Please read through the paper again, to correct format mistakes, e.g.:
· References should be numbered in order of appearance and indicated by a numeral or numerals in square brackets—e.g., [1] not [1]
· Please check the figure format and reference format.
Ø Is the mineralogical composition of the used materials (iron tailings, red mud, slag) and used fibers related to your results? A petrographic description or XRD analyses of the studied materials and fibers could be added.
Ø The results should be widely discussed. The section 3 must be rewritten.
Ø I think that the authors presented only investigations, but what about technology?
Ø The final question concerns for the economically aspects the process?
English is not bad and generally is easy to follow, but there are some evident grammar mistakes, which, in most cases, do not preclude the comprehension; hence the necessity of a revision, or not, would depend on the exigency of the journal in this aspect.
Author Response
We thank the reviewers for their comments and responses to this study. The authors have made changes in accordance with the reviewers' comments, which are described below:
1、Thanks to the reviewers for their comments; the author has added new references as required. (P17,L392-L396)
2、The author has re-explained the purpose of this experiment in the paper (including the revision of the abstract and introduction). (P1,P2)
3、Thank you for your reply; the author has added the literature referenced by the equipment method. (P5,L141-L142)
4、The author has added a table of abbreviations. (P16,L363)
5、Thank you for your suggestion; the reference format has been changed as required. (P17,P18)
6、The materials used (iron tailings, red mud, slag) are not related to the mineral composition of the fibers used, and the petrographic descriptions of some materials and fibers have been modified. (P12,L274-L290)
7、Thanks to the reviewers for their opinions; the author has adjusted and revised part of the content of the third section. (P9,P10,P11,P12)
8、Thank you for your comments. Based on previous research, the author adopts the DOE design method to study the fiber-reinforced controlled low-strength material (CLSM). This study compares the advantages and disadvantages of different types of fibers in CLSM, starting from the strength of CLSM after incorporation of fibers. In the concluding section, the prospect of this research is also added, including considering the incorporation of other solid waste materials to improve the application performance on the pavement base, exploring how to apply this material to higher-grade pavement, and conducting a more in-depth exploration of its working mechanism through the use of microscopic means, for example, XRD and FTIR. (P15,L338-L344)
9、The last question concerns the economic aspects of the process:
(1) Savings in material costs: The use of fibers mixed into concrete can reduce the use of traditional steel bars, thereby saving the purchase cost of steel bars. While the fiber itself is a cost, it is usually relatively inexpensive, so there are still some material cost savings overall.
(2) Reduction of construction costs: When fibers are mixed into concrete, cutting and binding of steel bars are not required, which saves related construction manpower and time costs. In addition, the fiber mixed with concrete can also be constructed by pumping, which improves construction efficiency and reduces the cost of manual handling.
(3) Energy saving and environmental protection: The incorporation of fiber into concrete can reduce energy consumption because a lot of energy is required in the process of steel bar manufacturing and installation. At the same time, the incorporation of fibers into concrete is also conducive to improving the durability and crack resistance of concrete, reducing the cost of repair and maintenance, and is also beneficial to the environment.
(4) Reduce the maintenance cost caused by concrete cracking: The incorporation of fiber into concrete can effectively control the cracking of concrete and reduce the maintenance cost caused by cracking, especially in some structures that are sensitive to cracking, such as ground slabs and pavement bases.

Reviewer 3 Report
The manuscript «Study on the influence of fiber on the mechanical properties of controllable low-strength materials» by Yafeng Qian and Mingyang Jiang was submitted for peer review.
I read the submitted manuscript with great interest. The authors turned to a very urgent problem: creation of material based on iron tailings with red mud, which is waste of aluminum production, and slag with fibers to increase strength of created material. The manuscript addresses an interesting topic that has potential for application in mining.
Despite of the actual topic and well-conducted study, the authors have failed to prove the relevance of the study. The manuscript has significant flaws that need to be corrected. Correction of the shortcomings listed below must be done to improve the quality of the manuscript, enhance the ease of perception of the presented material and increase the interest of a readers.
The use of various types of fibers in materials to control strength characteristics is not new. So, the scientist Patrikeyev (Russia) proposed the use of basalt fiber to increase the strength of concrete for the construction of a bridge across the river. Such concrete reduced the weight of the structure. Patent received. Patrikeyev developed his knowledge and used asphalt with basalt fiber on the same bridge. Patent received. Later, Professor Ermolovich (Russia) proposed basalt, asbestos, and steel (metal) fibers to increase the strength of the backfill for mining. Patents have been obtained for materials with basalt and asbestos fibers.
1.) From my point of view, this number of keywords is very few. In addition, keywords should be more direct and related to the content of the manuscript. Avoid abbreviations. Exceptions are established expressions, such as GPRS. Keywords enable the reader to quickly search for the necessary material and enable the author to popularize their research and increase interest and citations. But if this number of keywords satisfies the requirement of the journal, this comment is advisory.
2.) The abstract is not quite formed correctly. It is very blurry and framed incorrectly. It seems that the authors have taken certain phrases from the text and thus formed the abstract. The abstract should clearly indicate the purpose of the study, its importance for society (i.e. to characterize the problem), identify the methods and materials of the study, and the conclusions should be clearly and briefly formulated. There is no "starting point" in the abstract, that is, information about previous studies (one sentence is enough). From my point of view, in the abstract, such information begins with the statement: "Previously conducted studies have established that ...".
2.1) It is desirable to avoid narrative text in the abstract.
2.2) Try to use words and phrases: an analysis has been carried out; studied; developed; proposed; established and so on. It is advisable to start sentences in the abstract with these words and phrases.
2.3) At the end of the abstract, it is necessary to indicate the final result obtained by the authors, for example: A model has been developed that allows ...; A dependence has been established which is...; A pattern has been revealed...; An efficient system (technology) has been proposed, and so on.
The abstract should be revised.
3.) The manuscript has a sufficient list of references (40 references in total). But there is no comprehensive coverage of research in terms of geography of citations. No references to international studies in the field, especially on the work of Eastern European, Ukrainian, and Russian scientists. The list of references is intended to demonstrate the depth of the authors’ study of the material, the relevance and interest of their research.
3.1.) The depth of study is demonstrated with the number of references - is sufficient.
3.2.) Relevance – with the availability of research in recent years – is sufficient.
3.3.) Interest – with the availability of research by scientists from different countries - is not sufficient. I ask the authors to take this recommendation seriously. Since you are publishing your manuscript in an international publication, it is necessary to demonstrate the international relevance and interest of this issue. This can be done by analyzing the studies of scientists from different countries. It is imperative to supplement the list of references with studies of scientists from Eastern European countries over the past 3-5 years to show geographical (general/global) interest and relevance. Major revision of References might be sufficient if these tests have been performed. Otherwise, the paper should be considered as rejected in the present form. Below I present a few papers relevant to this study that could greatly improve the manuscript. The authors have the right to use the material proposed or offer their own versions of international studies to increase the geography of citation. The references must be supplemented.
4.) In the introduction when analyzing previous studies, the authors make inaccuracies or provide information that overloads the text and often their claims are not accompanied with evidence. It is important for readers to know the essence (main idea) of the research you are referring to when analyzing previous work. In the introduction, it is necessary to analyze the previously completed work and note what has been done, what are the shortcomings, and what has been done incorrectly. Such shortcomings are present throughout the Introduction. Authors need to revise the introduction, adjust, and supplement their statements with evidence.
4.1) I am not a native speaker, but nevertheless, in my opining, the authors form a very long sentences, which are very difficult to perceive. Such sentences greatly reduce the easy perception of the material.
4.2) In the introduction, the authors refer to several works and quite rightly state what is done in this study. However, the authors do not explain why this study is interesting: what has been done right or wrong, what can be learned from the study, what needs to be corrected or improved and why this research is important.
4.3) The authors use mining and processing waste to prepare the material. This is very correct from the point of view of preserving the environment. But the authors did not indicate this in the introduction. From my point of view, it is necessary to note the impact (accumulation) of waste on the environment and indicate the need for their disposal. One of such methods of disposal is the use of waste in road construction.
4.4) The authors in the introduction describe in sufficient detail the possibility of increasing the strength of the material with the help of various fibers. The analysis of research is carried out. However, the authors did not pay attention to other ways to control the characteristics of the material. It is necessary to eliminate this shortcoming.
5.) From my point of view, the authors abuse the names of scientists when mentioning the study, for example Wu Zhimin [14] and Ji Y [15]. A reference [14], [15] is sufficient. If the reader is interested in the name of the researcher, then it is easy to refer to the references list. It is important for the reader to know the essence (main idea) of the disclosed issue, not the name of the researcher.
6.) I would recommend avoiding group references, for example [17-20]. From my point of view, allowed up to three; more than three references are not acceptable and must be deciphered. Each paper you refer is unique and the studies you refer deserve more proper and careful review to demonstrate (and prove) its importance for the current research. It is necessary to demonstrate in detail the essence of each study and their need for your work. It has already been noted in recommendation (4.2) that you have many statements without indicating awareness. You will avoid group references by correcting this fact.
7.) At the end of the introduction, brief conclusion of the analytical study of earlier papers is absent. The authors did not summarize their analysis and did not identify unresolved issues. This conclusion should make it possible to characterize the actual question posed, the purpose of the study and the tasks to be solved to achieve this goal. For example: Analyzing the above, it can be noted that ... is a very topical issue. Therefore, the purpose of this study is ... and to achieve this, it is necessary to solve the following tasks: 1); 2); ... Such a conclusion allows the reader to understand the vector of the study, and the authors to correctly formulate the conclusions. It needs to be improved.
8.) Considering the comments (3) and (4), I would like to note that the authors have very poorly disclosed the main subject of the study. The impact of industrisl waste on the environment is quite large. Therefore, the issues of reducing this impact are relevant and scientists around the world are trying to minimize it. In addition, cement is a rather expensive component. And scientists are looking for an alternative or a way to increase the strength while reducing the cement binder. In recent years, many studies have been carried out on the study of a material based on waste. For example:
8.1) Ermolovich, E.A.; Ivannikov, A.L.; Khayrutdinov, M.M.; Kongar-Syuryun, C.B.; Tyulyaeva, Y.S. Creation of a Nanomodified Backfill Based on the Waste from Enrichment of Water-Soluble Ores. Materials 2022, 15(10), 3689. https://doi.org/10.3390/ma15103689. This work is similar in terms of goals, objectives and research methods to the manuscript submitted for peer review. The authors study a composite based on waste from the processing of water-soluble ores. To increase the strength characteristics of the created material, fullerene-astarlene is used as a nanomodified additive. From my point of view, this work should be used in the analysis of previously performed studies, since it uses the methods of mechanical, microstructural, X-ray phase and petrographic analyzes to confirm arguments.
8.2) Kongar-Syuryun, Ch.B.; Faradzhov, V.V.; Tyulyaeva, Yu.S.; Khayrutdinov, A.M. Effect of activating treatment of halite flotation waste in backfill mixture preparation. Mining Informational and Analytical Bulletin 2021, 2021(1), 43–57. https://doi.org/10.25018/0236-1493-2021-1-0-43-57.
8.3) Khayrutdinov, A.; Kongar-Syuryun, Ch.; Kowalik, T.; Faradzhov, V. Improvement of the backfilling characteristics by activation of halite enrichment waste for non-waste geotechnology. IOP Conf. Ser.: Mater. Sci. Eng. 2020, 867(1), 012018. https://doi.org/10.1088/1757-899X/867/1/012018. Papers (8.2) and (8.3) suggest activation treatment of tailings before mixing to improve the strength and rheological characteristics. Activation treatment or activating additive is one of the ways to improve the quality of the created material. From my point of view, the studies (8.2) and (8.2) will suit the authors in the analysis of previously completed works to demonstrate various options for controlling the characteristics of the created material. In this way the authors will avoid shortcomings in recommendation (4.4).
If the authors become familiar with the works presented in (8.1), (8.2), (8.3) they will be able to properly form the introduction, enrich their manuscript with international research by scientists from Poland, Czech Republic, Slovenia, Slovakia, Russia, Germany and demonstrate the depth of their material, as well as eliminate the remarks (3) and (4).
9) Of particular interest to me, and I think readers as well, is SEM and X-RAY analysis for studying composition of tails, red mud, and slags. From my point of view, the next point should be indicated:
9.1) equipment used for studies (brand/model);
9.2) database for X-RAY analysis;
9.3) Method of preparation of samples;
9.3.1) whether grinding was carried out, if so, to what fraction (only red sludge is indicated);
9.3.2) whether drying was carried out, if so, what the initial and final moisture content was (only red sludge is indicated);
9.3.3) equipment for drying;
10.) Method for determining density of materials: tails, red mud, slag.
11.) Type of tails: current or stale tails.
12.) Description of experiment for creating of material with following information:
12.1) A standard composition (with a cement binder). If there was no such standard, the reason for this should be stated;
12.2) how was the convergence of the results achieved;
12.3) how was the homogenization (mixing) carried out; what is the mixing tool; what is the mixing velocity and time;
12.4.) what is the sequence of filling of the components
12.5) how the homogeneity of the composition (thoroughness of mixing) was achieved, provided that the amount of some components (for example, fibre) in the composite is minimal;
12.6) composition (recipe): binder/ aggregate/activator (fibres)/ grouting fluid 11.10) how many samples were prepared. To eliminate remarks (8) – (9), I would recommend reading the work (8.1). The recommended paper is similar to the one submitted for peer review. The article (8.2) describes the methodology in sufficient details.
13.) Further vector (direction) of research can be indicated in Discussion section.
14.) Conclusion is not correctly formed. Conclusion – summary of the study without repeating the wording given earlier in the manuscript. It is exactly the way of presenting the material that makes it easier for the reader to perceive the information presented. The mistake of incorrectly forming conclusion is a consequence of the incorrect presentation of the introduction noted by me in remark (6) due to the fact that when writing the introduction, the aims and objectives are not formulated. Conclusions are overloaded with information that should be placed in other section. For example, first paragraph refers to Materials and Methods section. The authors indicated this information in relevant section, so it is excessive in Conclusion. The next information refers to Results and Discussion section. The information was indicated earlier too.
Conclusions should briefly characterize the result of the study, for example:
(1) the strength of the created material demonstrated a directly/inversely proportional dependence on the fiber size.
(2) the dependence of … was obtained.
(3) it was found that ...
(4) and so on. The conclusion needs to be revised.
Summary: The manuscript is not a finished research work. The corrections are needed. The chosen research topic is relevant. From my point of view, the authors failed to present their research correctly and clearly, which reduced its value and worsened the ease of perception of the material presented. From my point of view, the manuscript cannot be published in the open press without correction in accordance with my suggestions.
Author Response
We thank the reviewers for their comments and responses to this study. The authors have made changes in accordance with the reviewers' comments, which are described below:
1、Thank you for your reply; the author has modified the keywords. (P1,L21)
2、The author has modified the format of the abstract according to the reviewer's suggestion. (P1,L6-L20)
3、The author has added the work of Eastern European, Ukrainian, and Russian scientists (the references given by the reviewers in paragraph 8 have been adopted. (P18,L421-L422;P17,L372-L373;P17,L376-L377)
4、Thank you for your reply. The author has partially revised the introduction and summarized the literature. (P1,P2,L45)
5、Thank you for your suggestion; the author has made revisions (including group citations) as requested by the reviewers. (P1,L40-L41;)
6、Amendments have been made, and some documents have been changed according to the requirements of the reviewers. (P17,P18)
7、The author has added an analysis and summary of the references. (P2,L64-L73)
8、Thanks to the reviewers for their suggestions; the author has revised and added relevant literature as required. (P18,L421-L422;P17,L372-L373;P17,L376-L377)
9、The equipment models of SEM and XRF have been mentioned in Table 3, and the XRF test is carried out by relevant institutions. Sample preparation methods are indicated in the text. (P6,L151-L162).The drying equipment and method of red mud are also explained in the text. (P3,L94-L108).However, the author did not test the original water content of red mud.
10、The density test method of raw materials (including tailings, red mud, and slag) has been explained in the text. (P2,L83)
11、The tailings used are waste tailings after mining.
12、Thank you for your reply. The author has added the experimental description of the material fabrication, including the number of samples required for the test and the reference specification. (P6,L151-L162,L169-L172)
13、In this study, fibers are added to CLSM and applied to the pavement base, which can provide a more stable and durable pavement structure in the application of the pavement base, reduce the cost of maintenance and repair, and at the same time improve the safety and reliability of road traffic, also having a positive impact. Fiber-reinforced CLSM has broad application prospects on pavement bases, especially in projects that require rapid construction and improved pavement performance.
14、Thank you for the reviewer's reply; we have revised the conclusion part according to the reviewer's request. (P14,L319-L332;P15,L338-L344)

Reviewer 4 Report
The authors study the influence of different fibre types on mechaniczl properties of low strength matefial. The . The relevance and importance of the research is explained by hudge ammount of waste products that can be utiljzed using the propksed methkdology. The topjc is very intefesting for the readers, but the authors should consider transferring the paper to a special issue dealing with re ycling and using waste products for producing construction materials.
A similar research was performed by L. Dvorkin et al for regular concrete. It would be interesting to compare the results and to explain the readers what is different in the proposed approach. Dvorkin et al. used mathematical experiment planning for investigating the efficiency of different fibers. Is the method suitable also for lkw-strength materials?
Author Response
We thank the reviewers for their comments and responses to this study. The authors have made changes in accordance with the reviewers' comments, which are described below:
Thank you for the reviewer's reply! The author has redesigned and revised parts of the article, including the abstract, introduction, sample preparation, and analysis of experimental results. Additionally, we carefully read the relevant literature mentioned by the reviewers and referred to the relevant research of L. Dvorkin in our article(P17,L382-L383). The author believes that L. Dvorkin's article provides valuable insights as it starts from a mathematical perspective and utilizes the response surface design method to analyze the composition of fiber concrete and optimize the design. This aspect is worthy of our reference and consideration.
In this paper, building upon previous studies, the author controls the cited indicators within a certain range and employs the DOE Taguchi design to optimize the design. The DOE design offers advantages such as shorter test time, stronger reliability, and lower cost, which have also been highlighted in the aforementioned article(P7,L181-L183). These reasons primarily contribute to our decision to adopt the DOE design methodology.

Round 2
Reviewer 2 Report
The authors carefully followed the comments and suggestions, made appropriate corrections and the manuscript in the present form was sufficiently improved with respect to the previous version. I recommend to accept this manuscript for publication.
Reviewer 3 Report
Comments and Suggestions for Authors
As can be seen from the submitted manuscript and the explanatory note to the review, the authors did a lot of work to
make changes in accordance with the comments. The revised manuscript is a completed scientific study on a highly
relevant topic: creation of material based on iron tailings with red mud, which is waste of aluminum production, and
slag with fibers to increase strength of created material. The revised version of the manuscript, in my opinion, fully satisfies the requirements of a scientific article and can be published in the open press.